# Prevalence and associated factors of adolescent fatherhood in Ethiopia: A multilevel analysis using the 2016 Ethiopian demographic health survey data

**Misganaw Gebrie Worku**[1]*, **Getayeneh Antehunegn Tesema**[2], **Achamyeleh Birhanu Teshale**[2]

**1** Department of Human Anatomy, College of Medicine and Health Science, School of Medicine, University of Gondar, Gondar, Ethiopia, **2** Department of Epidemiology and Biostatistics, Institute of Public Health, College of Medicine and Health Sciences, University of Gondar, Gondar, Ethiopia

* misgeb2008@gmail.com

## Abstract

### Background

Though the consequences of teenage pregnancy and early motherhood has been studied very well, little is known about the magnitude as well as the determinants of adolescent fatherhood. Unlike adolescent motherhood, limited public health programs are working on adolescent fatherhood. Currently, in developed countries, there is an increased work to acknowledge teen fathers in clinical practice and in the research forum, but still, there are more issues that need to be addressed in developing countries including Ethiopia. Therefore, this study aimed to investigate the prevalence and associated factors of adolescent fatherhood in Ethiopia based on the nationally representative survey.

### Methods

This study used the 2016 Ethiopian Demographic and Health Survey (EDHS) data. A total weighted sample of 4455 adolescent men was included for the final analysis. For the associated factors, multilevel logistic regression analysis was conducted to consider the hierarchical nature of the EDHS data. Intra-class Correlation Coefficient (ICC), and deviance (-2LLR) were used for model comparison and for assessing model fitness. The model with the largest adjusted $R^2$, lowest Bayesian Information Criteria (BIC) and the smallest cross-validation prediction error were considered as the best-fitted model. In the multivariable analysis, the Adjusted Odds Ratio (AOR) with 95% Confidence Interval (CI) were reported to declare the presence of statistically significant factors associated with adolescent fatherhood, and variables with p-value <0.05 were considered as a significant variable.

### Results

The prevalence of adolescent fatherhood in Ethiopia was 6.79% [95%CI; 6.08%, 7.56%]. Adolescent men with contraceptive knowledge [AOR = 4.25; 95%CI = 1.23, 14.69], age in

**Data Availability Statement:** The 2016 EDHS data used in this study are third party data from the Demographic and Health Surveys Program website (https://dhsprogram.com/data/available-datasets.

cfm) and can be accessed following the protocol outlined in the Methods section.

**Funding:** The authors received no specific funding for this work.

**Competing interests:** The authors have declared that no competing interests exist.

**Abbreviations:** AOR, Adjusted Odds Ratio; COR, Crude Odds Ratio; CI, Confidence Interval; CSA, Central Statistical Agency; DHS, Demographic Health Survey; EA, Enumeration Area; EDHS, Ethiopian Demographic Health Survey; ICC, Intraclaster correlation; LLR, likelihood Ratio; MOR, Median Odds Ratio; PCV, Proportional Change in Variance; SNNPR, Southern Nation Nationality, and Peoples Region.

20 to 24 years [AOR = 7.93; 95%CI = 3.66, 17.27] and being Muslim [AOR = 1.84; 95%CI = 1.02, 3.39] were significantly associated with Higher odds of adolescent fatherhood. Individuals who initiate sex lately [AOR = 0.35; 95%CI = 0.22, 0.54], being in female-headed family [AOR = 0.46; 95%CI; 0.26, 0.82] and being from Amhara region [AOR = 0.35; 95%CI = 0.14, 0.84] were significantly associated with lower odds of adolescent fatherhood.

## Conclusion

In this study, adolescent fatherhood was a common public health problem in Ethiopia as it is closely linked with poor quality of life and premature death (year of potential life lost). Age of respondent, knowledge of respondent about contraceptive methods, early initiation of sex, religion, sex of household head, and region were significantly associated with adolescent fatherhood. Therefore, program planners and decision-makers should give special attention to adolescent men through enhancing reproductive health services utilization and their knowledge towards it to decrease the incidence of adolescent fatherhood.

## Background

Adolescent fatherhood is defined as a young male under his 24[th] birthday who is taking responsibility for the procreation of an offspring regardless of the age of the woman [1, 2]. The challenges of teen pregnancy and motherhood have been considered in-depth, but attention to adolescent fatherhood has been far less conspicuous [3, 4]. Although many programs related to adolescent motherhood are available, programs related to adolescent fatherhood are limited [5]. Recently, there is an increased work to acknowledge teen fathers in clinical practice and the research forum, but still, there are more needs to be addressed [4, 6]. In developing countries, there are more than one billion adolescents who are physically old enough to reproduce themselves but too young to be responsible for their partner and children [1]. Teenage fathers are relatively absent from public statistics and no study provides information about the appropriate age for fatherhood, unlike a voluminous study on motherhood and female fertility [1]. While teen pregnancy is studied broadly worldwide, the acquiescence of many communities and cultures towards teen fatherhood is 'appalling in developing countries especially in sub-Saharan Africa [1]. Even though there are gaps in the literature concerned about adolescent fatherhood, studies done in sub-Saharan Africa reported that the prevalence of adolescent fatherhood ranges from 5.9% to 13.5% [4, 7]. Several factors are correlated with becoming a father during adolescence, including low income and poor academic achievement, being old aged, early sexual initiation, knowledge of contraceptive, condom use, and religion [4, 7].

Young fathers are more likely to have economic and employment challenges and are more often economically disadvantaged than adult fathers [8–10]. Teen fathers are more likely to live in deprived areas and suffer unemployment and lack of access to healthcare services [11]. Although there is increased interest in adolescent fathers, the knowledge base needed by social work practitioners who provide services to adolescent parents has not expanded [11]. Adolescent fatherhood had a huge impact on himself, his offspring, and his partner. Also, adolescent fatherhood had a negative impact on their educational achievement, the economic burden to themselves and the community, and social health [12].

Despite the above-mentioned impacts of adolescent fatherhood, previous researches had focused primarily on the experience of a young mother (teenage pregnancy) and until recently

little attention has been paid to teenage fatherhood [13]. Besides, reducing adolescent fatherhood is closely connected to both promoting responsible fatherhood and reduction in teenage pregnancy [1]. Therefore, this study aimed to assess the prevalence and associated factors of adolescent fatherhood in Ethiopia. The findings of this study could give an insight/input for policymakers, as well as other governmental and non-governmental organizations for taking appropriate interventions to reduce the incidence of adolescent fatherhood and its consequences.

## Methods

### Study area

The study was conducted based on the 2016 EDHS data in Ethiopia which is found in the horn of Africa. The country encompasses 1.1 million sq. Km and has a large geographical diversity ranging from 4550 meters above sea level to 110 meters below sea level. It has nine regional states (Tigray, Afar, Amhara, Oromia, Somali, Benishangul-Gumuz, Southern Nations Nationalities and People Region (SNNPR), Gambela and Harari regions) and two city administrations (Addis Ababa and Dire Dawa) which is again subdivided into 68 zones, 817 districts and 16,253 kebeles (the country's lowest administrative units) in the country's administrative structure.

### Data source and sampling procedure

Secondary data analysis was conducted based on the 2016 EDHS data [14]. EDHS 2016 was the fourth survey conducted in every five-year interval to generate updated health and health-related indicators. In EDHS, a two-stage stratified cluster sampling technique was employed to select the participants. In the first stage, a total of 645 enumeration areas (EAs) (202 in urban areas and 443 in rural areas) were selected using the 2007 Population and Housing Census (PHC) as a sampling frame, with a probability proportional to the EA scale. In the second stage, a fixed number of 28 households per EAs was selected. In this survey, a total of 16650 households, 12688 men, and 15683 women were interviewed successfully. For this study, the Men's Record (MR) data set was used, and a total weighted sample of 4455 men aged 15 to 24 years was included for the analysis. The detailed sampling procedure is presented in the EDHS 2016 report [14].

### Study variables

**Dependent variable.** The dependent variable was adolescent fatherhood (ever had of at least a child before the 24th birthday). It was generated from the EDHS variables "the number of children ever fathered" which was recorded as no "0" if he never had a child, and yes "1" if he had one or more children.

**Independent variable.** For this study both individual and community-level variables were included as independent variables. The individual-level variables considered for this study were; the age of respondent, educational level, religion, number (frequency) of unions, occupational status, wealth status, sex of household head, knowledge about contraceptive methods, and age at first sex. Residence and region were considered as the community-level variables (Table 1).

**Data management and analysis.** Data extraction, recoding and analysis (both descriptive and analytical) were done using STATA version 14 statistical software. The weighted data were used for analysis to get a reliable estimate and standard error. Descriptive statistics presented summary statistics such as proportion and median. Since, the DHS data has a hierarchical

**Table 1. Description and measurement of independent variables.**

| Independent variables and their description/categorization | |
| --- | --- |
| Individual-level variables | |
| Age group | current age of the men and re-coded in to two categories with values of "0" for 15–19, "1" for 20–24. |
| Religion | Re-coded in four categories with a value of "1" for Orthodox, "2" for Muslim, "3" for protestant, and "4" for other religious groups (combining catholic, traditional and the other religious categories as most young men in this category are small in number). |
| Wealth Index | It was coded as "poorest", "poorer", "Middle", "Richer", and "Richest in the EDHS data set." For this study we recoded it in to three categories as "poor" (includes the poorest and the poorer categories), "middle", and "rich" (includes the richer and the richest categories) |
| Occupation | Re-coded in five categories with a value of "0" for unemployed, "1" for professional, "2" for clerical/sales/services, "3" for farming/unskilled and "4" for skilled manual. |
| Age at 1st sex | Recoded in two categories with a value of "0" for age≤19 and "1" for age 20–24. |
| Frequency of union | The variable frequency (number) of union was recorded as once and more than once in the dataset and we use it without change. |
| Educational status | This is the minimum educational level adolescent man achieved and coded in to four groups with a value of "0" for no education, "1" for primary education, "2" for secondary, and "3" for higher education in the data set. |
| Sex of household | The variable sex of household head was recorded as male and female in the dataset and we used without change. |
| Contraceptive knowledge | Recoded in to two categories with value of 0 for "no" if adolescent man don't know any of the contraceptive methods and 1 for "Yes" if a man know any (traditional contraceptive method and/or modern contraceptive method) of the contraceptive methods. |
| Community level variables | |
| Type of place of residence | The variable place of residence was recorded as rural and urban in the dataset and used was used without change for this study. |
| Region | The variable region was coded in to 11 categories in the dataset and we were retained without change. |

structure, which violates the independent assumptions of the standard logistic regression model, a multilevel logistic regression analysis was implemented. To assess whether there was a significant clustering effect or not, the Intra-class Correlation Coefficient (ICC) and the Median odds Ratio (MOR) were done and it indicates the presence of a statistically significant clustering effect that should be considered during analysis using advanced statistical models. Therefore, a multilevel binary logistic regression analysis was employed to investigate the statistically significant individual level and community level variables associated with adolescent fatherhood. Multi-collinearity between independent variables was assessed using Variance Inflation Factor (VIF) and Tolerance by running pseudo linear regression analysis as VIF is dependent on the coefficient of determination, and the mean VIF was less than 5. Four models (null model; a model containing only the outcome variable, model I; a model containing individual-level variables, II; a model containing community-level variables, and the final model (model III); a model which contain both individual and community level variables) were constructed, and model comparison was done based on deviance (-2LLR) as the models were nested models. The final model (model III) was the best-fitted model since it had the lowest deviance value. Besides, Proportional Change in Variance (PCV) was done to assess by how much the final model explains the variability in relative to the null model. The lasso method was used for variable selection and model prediction. The model with the largest adjusted $R^2$, lowest Bayesian Information Criteria (BIC) and the smallest cross-validation prediction error were considered as the best-fitted model. In the multivariable multilevel analysis, the Adjusted Odds Ratio (AOR) with the 95% Confidence Interval (CI) were reported to declare the significantly associated factors of adolescent fatherhood.

**Ethical consideration.**   Since the study was a secondary data analysis of publically available survey data from the MEASURE DHS program, ethical approval and participant consent were not necessary for this particular study. We requested DHS Program and permission was granted to download and use the data for this study from http://www.dhsprogram.com. The Institution Review Board approved procedures for DHS public-use datasets do not in any way allow respondents, households, or sample communities to be identified. There are no names of individuals or household addresses in the data files. The document was submitted to the university of Gondar ethical review board and the ethical review board approved that ethical clearance is not needed for such type of study, since it is based on nationally representative EDHS data.

## Results

### Socio-demographic characteristics of study participants

From a total of 4455 respondents, more than half (57.73%) of adolescents were aged 15-19 years. The majority (92.13%) of adolescent men started their sexual activity early, and about 61.59% of the participants had attained primary education. Regarding contraceptive knowledge, the majority (96.42%) of respondents had knowledge about the different contraceptive methods and 88.55% of subjects had a union of only once. Nearly half (45.06%) of young men were orthodox religious followers, and the majority (67.63%) of study participants were involved in farming or unskilled manual works. More than three-fourth (80.53%) of the respondents were rural residents, and about 37.2% were living in the Oromia region (Table 2).

### Prevalence of adolescent fatherhood in Ethiopia

The prevalence of adolescent fatherhood in Ethiopia was 6.79% [95%CI = 6.08%, 7.56%]. It ranges from 3.05% in Addis Ababa to 17.79% in the Oromia region (Fig 1).

### Random effect analysis results

The random effect model of a multilevel analysis was assessed using ICC, MOR, and PCV. In the null model the ICC value was 0.28, indicates that 28% of the total variation on adolescent fatherhood was attributable to the differences across clusters while the remaining 72% of the total variation on adolescent fatherhood was attributable to the between individual differences. The MOR value in the null model was 2.96 indicates that there was significant variation in the odds of experiencing adolescent fatherhood across clusters. The PCV valve in the final model (model III) was 0.96, indicates that about 96% of the variation in adolescent fatherhood was explained by both individual and community-level factors. Regarding model fitness, the final model (model III), which incorporates both individual and community level factors, was the best-fitted model since it had the lowest deviance (628.490) (Table 3).

### Factors associated with adolescent fatherhood

In the multivariable multilevel binary logistic regression analysis, knowledge on contraceptive methods, age of respondent, age at first sex, religion, sex of household head, and region were significantly associated with adolescent fatherhood.

The odds of adolescent fatherhood among men with contraceptive knowledge were 4.25 [AOR = 4.25; 95%CI = 1.23, 14.69] times higher compared to their counterparts. Regarding respondent age, men aged 20 to 24 years had 7.93 [AOR = 7.93; 95%CI = 3.66, 17.27] times higher odds of being an adolescent father than those aged 15-19 years. The odds of being adolescent fatherhood among young men who initiate sex lately were decreased by 65%

**Table 2. Sociodemographic characteristics of the respondents.**

| Variables | | Frequency (%) |
|---|---|---|
| Respondent age | 15–19 | 2572(57.73%) |
| | 20–24 | 1883(42.27%) |
| Age at 1st sex | ≤19 | 4104(92.13%) |
| | 20–24 | 351(7.87%) |
| Educational level | No education | 543(12.18%) |
| | Primary education | 2744(61.59%) |
| | Secondary education | 910(20.43%) |
| | Higher education | 258(5.8%) |
| Knowledge of contraceptive | Yes | 4296(96.42%) |
| | No | 159(3.58%) |
| Frequency of union | Once | 501(88.55%) |
| | More than once | 65(11.45%) |
| Religion | Orthodox | 2007(45.06%) |
| | Muslim | 1339(30.05%) |
| | Protestant | 1013(22.745) |
| | Other | 96(2.16%) |
| Wealth status | Poor | 1424(31.97%) |
| | Middle | 846(19%) |
| | Rich | 2184(49.03%) |
| Occupation | Unemployed | 811(18.20%) |
| | Professional | 85(1.92%) |
| | clerical/service/sales | 314(7.06%) |
| | Farming/unskilled | 3013(67.63%) |
| | Skilled | 23(5.19%) |
| Region | Tigray | 310(6.96%) |
| | Afar | 29(0.65%) |
| | Amhara | 1130(25.36%) |
| | Oromia | 1657(37.2%) |
| | Somalia | 42(0.94%) |
| | Beni shangul | 41.73(0.94%) |
| | SNNPR | 916(20.56%) |
| | Gambella | 15(0.33%) |
| | Harari | 10(0.22) |
| | Adiss Ababa | 195(4.38%) |
| | Dire Dawa | 27(0.6%) |
| Residence | Urban | 867(19.47%) |
| | Rural | 3588(80.53%) |
| Sex of household head | Male | 347(77.92%) |
| | Female | 984(22.08%) |

[AOR = 0.35; 95%CI = 0.23, 0.54] than those who initiate sex early. Being Muslim had 1.84 [AOR = 1.84; 95%CI = 1.02, 3.39] times higher odds of being a father at adolescent age compared to Orthodox Christian religion follower. The odds of being adolescent fatherhood among men from the female-headed household were decreased by 54% [AOR = 0.46; 95%CI; 0.26, 0.82] compared to those from male-headed households. The odds of adolescent fatherhood among adolescents in the Amhara region were decreased by 65% [AOR = 0.35; 95% CI = 0.14, 0.84] compared with men in the Tigray region (Table 4).

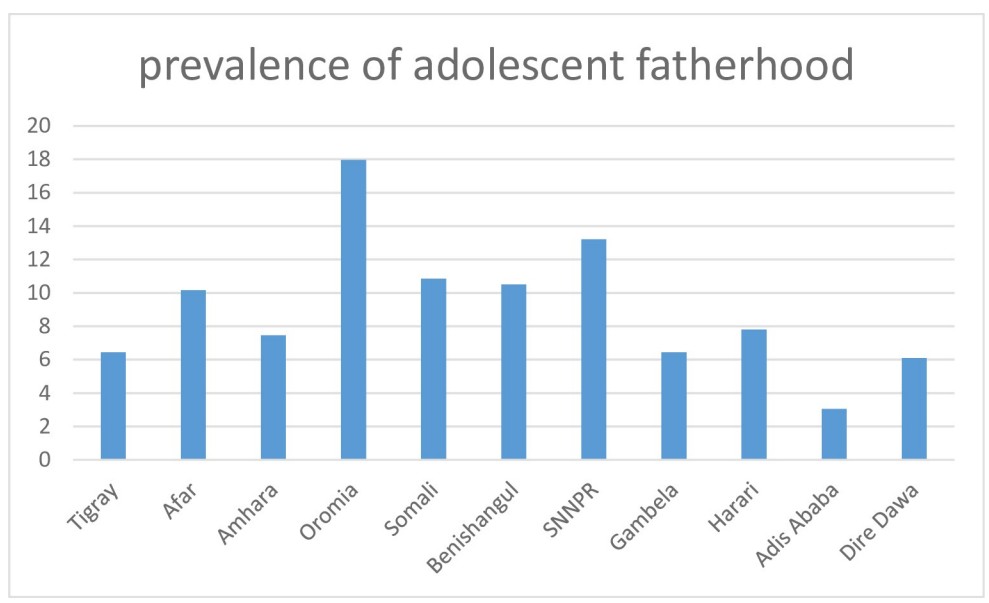

**Fig 1. Prevalence of adolescent fatherhood by region.**

## Discussion

In this study knowing contraceptive methods, age of respondent, age at first sex, Muslim religion follower, being from a female-headed family and those from the Amhara region were significant factors associated with adolescent fatherhood. Muslim religion followers had higher odds of being adolescent fatherhood, which is, in contrast, to a study done in sub-Saharan Africa [7]. This might be associated with the variation in religious doctrine, law, and practice towards early marriage [15, 16]. Besides, the religious difference might be associated with the difference in the socio-economic and cultural variation of the study participants as commonly Muslim religious followers didn't use family planning methods, they may have a child early than others.

Respondents with early initiation of sex had more chance of being a father at an early age as compared to their counterparts, which is in agreement with the study done in Brazil [17]. This might be associated with individuals at an earlier age who might not be matured in knowledge of contraceptives and also can't afford it because of economical dependency. In our study adolescents in the old age group had a higher chance of being a father, which is contrary to the study conducted in sub-Saharan Africa [7]. This might be associated with as age increases,

**Table 3. Random effect model and model fitness for the assessment of adolescent fatherhood in Ethiopia.**

| Parameter | Null model | Model I | Model II | Model III |
|---|---|---|---|---|
| Community-level variance | 1.31 | 0.07 | 0.89 | 0.043 |
| ICC | 0.28 | 0.02 | 0.212 | 0.012 |
| MOR | 2.96 | 0.68 | 2.45 | 0.53 |
| PCV | Reff | 0.94 | 0.32 | 0.96 |
| **Model fitness** | | | | |
| Log likelihood | -1058.4807 | -323.98125 | -1026.2411 | -319.245 |
| Deviance | 2116.9614 | 646.96250 | 2052.4822 | 628.490 |

**Table 4. Multilevel analysis for the assessment of determinants of adolescent fatherhood in Ethiopia, 2016.**

| Variables | | Adolescent Fatherhood | | Crude Odds Ratio(95%CI) | Adjust Odds Ratio (95%CI) |
|---|---|---|---|---|---|
| | | Yes | No | | |
| Knowledge of contraceptive | Yes | 299 | 3997 | 3.26(1.33, 7.95) | 4.25(1.23, 14.69)* |
| | No | 4 | 156 | 1 | 1 |
| Number of union | Once | 266 | 235 | 1 | 1 |
| | More than once | 32 | 33 | 1.79(0.94, 3.39) | 1.33(0.70, 2.51) |
| Respondent age | 15–19 | 10 | 2562 | 1 | 1 |
| | 20–24 | 292 | 1591 | 43.20(23.96, 78.24) | 7.93(3.66, 17.27)* |
| Age at 1st sex | ≤19 | 203 | 3901 | 1 | 1 |
| | 20–24 | 100 | 251 | 7.48(5.29, 10.57) | 0.35(0.22, 0.54)* |
| wealth status | Poor | 169 | 1255 | 1 | 1 |
| | Middle | 54 | 792 | 0.69(0.48, 1.00) | 085(0.48, 1.51) |
| | Rich | 78 | 2106 | 0.32(0.23, 0.44) | 0.59(0.34, 1.04) |
| Educational level | No education | 72 | 471 | 1 | 1 |
| | Primary education | 188 | 2556 | 0.41(0.29, 0.58) | 0.86(0.51, 1.45) |
| | Secondary education | 37 | 873 | 0.34(0.22, 0.52) | 0.67(0.33, 1.32) |
| | Higher education | 6 | 252 | 0.29(0.15, 0.53) | 0.5(0.19, 1.29) |
| Religion | Orthodox | 71 | 1936 | 1 | 1 |
| | Muslim | 150 | 1188 | 3.20(2.25, 4.56) | 1.86(1.02, 3.39)* |
| | Protestant | 73 | 940 | 2.21(1.43, 3.40) | 1.69(0.82, 3.49) |
| | Others | 7 | 89 | 1.70(0.61, 4.76) | 6.16(0.57, 66.13) |
| Region | Tigray | 11 | 299 | 1 | 1 |
| | Afar | 4 | 25 | 4.15(1.90, 9.02) | 0.91(0.31, 2.62) |
| | Amhara | 41 | 1089 | 1.03(0.47, 2.25) | 0.35(0.14, 0.84)* |
| | Oromia | 162 | 1495 | 2.72(1.35, 5.46) | 1.59(0.61, 4.12) |
| | Somalia | 10 | 114 | 2.81(1.33, 5.94) | 1.65(0.51, 5.28) |
| | Beni shangul | 4 | 38 | 3.15(1.46, 6.78) | 1.07(0.41, 2.76) |
| | SNNPR | 63 | 853 | 2.20(1.07, 4.52) | 1.56(0.56, 4.35) |
| | Gambella | 1 | 14 | 1.50(0.65, 3.46) | 0.73(0.23, 2.31) |
| | Harari | 2 | 9 | 3.19(1.41, 7.22) | 1.86(0.56, 6.13) |
| | Adiss Ababa | 4 | 191 | 0.61(0.23, 1.59) | 0.71(0.18, 2.82) |
| | Dire dawa | 2 | 25 | 1.47(0.63, 3.41) | 1.01(0.32, 3.12) |
| Residence | Urban | 24 | 844 | 1 | 1 |
| | Rural | 279 | 3308 | 2.96(1.98, 4.44) | 0.49(0.23, 1.06) |
| Sex of household head | Male | 286 | 3185 | 1 | 1 |
| | Female | 17 | 967 | 0.36(0.24, 0.53) | 0.46(0.26, 0.82)* |

*p-value<0.05; AOR: Adjusted Odds Ratio, CI: Confidence Interval, COR: Crude Odds Ratio.

teenagers will have more exposure to sex and their chance of being married which in turn increases their chance of having children [18].

We also found that adolescent men from a female-headed family had lower odds of being a father at an early age. This might be due to females had more knowledge about contraceptive methods and utilize the methods appropriately, which in turn prevent adolescent fatherhood [19].

Surprisingly, an adolescent with knowledge of contraceptive methods had a higher chance of being an adolescent father. This might be associated with even though an adolescent father had good knowledge of contraceptive methods, he may be poor concerning the

implementation strategies [20]. Also, it might be associated with a higher rate of contraceptive failure because of inappropriate use like unskilled and inconsistent use of condom.

Adolescent men from the Amhara region had lower odds of being a father early. This regional variation might be associated with the variation in educational achievement and socioeconomic status.

### Strength and limitation of the study

The strength of this study was since it was based on weighted nationally representative data with large sample size. The other strength was also we used an appropriate statistical approach to accommodate the hierarchical nature of the data. Moreover, since it is based on the national survey data the study has the potential to give insight for policy-makers and program planners to design appropriate intervention strategies both at national and regional levels. However, this study had limitations in that the EDHS survey was based on respondents' self-report and might have the possibility of recall bias. Furthermore, in EDHS variables such as community attitude towards marriage, norms, values, and religious beliefs towards marriage and father-hood were not collected even if these variables are important variables that influence adolescent fatherhood.

### Conclusion

In this study, the prevalence of adolescent fatherhood was relatively higher. In the multivariable multilevel analysis of adolescent men with contraceptive knowledge, those aged from 20 to 24 years and being Muslim had higher odds of adolescent fatherhood. While, individuals who initiate sex lately, those from the female-headed family and being from the Amhara region had lower odds of being a father at an early age. Therefore, program planners and decision-makers should give special attention to high-risk groups to decrease adolescent fatherhood.

### Acknowledgments

We greatly acknowledge MEASURE DHS for granting access to the Ethiopia Demographic and Health Surveys data.

### Author Contributions

**Conceptualization:** Misganaw Gebrie Worku, Getayeneh Antehunegn Tesema, Achamyeleh Birhanu Teshale.

**Data curation:** Misganaw Gebrie Worku, Getayeneh Antehunegn Tesema, Achamyeleh Birhanu Teshale.

**Formal analysis:** Misganaw Gebrie Worku, Getayeneh Antehunegn Tesema, Achamyeleh Birhanu Teshale.

**Funding acquisition:** Misganaw Gebrie Worku, Getayeneh Antehunegn Tesema, Achamyeleh Birhanu Teshale.

**Investigation:** Misganaw Gebrie Worku, Getayeneh Antehunegn Tesema, Achamyeleh Birhanu Teshale.

**Methodology:** Misganaw Gebrie Worku, Getayeneh Antehunegn Tesema, Achamyeleh Birhanu Teshale.

**Project administration:** Misganaw Gebrie Worku, Getayeneh Antehunegn Tesema, Achamyeleh Birhanu Teshale.

**Resources:** Misganaw Gebrie Worku, Getayeneh Antehunegn Tesema, Achamyeleh Birhanu Teshale.

**Software:** Misganaw Gebrie Worku, Getayeneh Antehunegn Tesema, Achamyeleh Birhanu Teshale.

**Supervision:** Misganaw Gebrie Worku, Getayeneh Antehunegn Tesema, Achamyeleh Birhanu Teshale.

**Validation:** Misganaw Gebrie Worku, Getayeneh Antehunegn Tesema, Achamyeleh Birhanu Teshale.

**Visualization:** Misganaw Gebrie Worku, Getayeneh Antehunegn Tesema, Achamyeleh Birhanu Teshale.

**Writing – original draft:** Misganaw Gebrie Worku, Getayeneh Antehunegn Tesema, Achamyeleh Birhanu Teshale.

**Writing – review & editing:** Misganaw Gebrie Worku, Getayeneh Antehunegn Tesema, Achamyeleh Birhanu Teshale.

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
