## [Decision Letter · Decision Letter 0]

15 Sep 2020

PONE-D-20-22844

Prevalence and associated factors of adolescent fatherhood in Ethiopia: a multilevel analysis using EDHS 2016 data.

PLOS ONE

Dear Dr. Misganaw Gebrie Worku,

Thank you for submitting your manuscript to PLOS ONE. After careful consideration, we feel that it has merit but does not fully meet PLOS ONE’s publication criteria as it currently stands. Therefore, we invite you to submit a revised version of the manuscript that addresses the points raised during the review process.

We look forward to receiving your revised manuscript.

Kind regards,

Yuka Kotozaki

Academic Editor

PLOS ONE

Journal Requirements:

2.

Thank you for stating the following in your Competing Interests section: 

"none"

3.

We note that you have indicated that data from this study are available upon request. PLOS only allows data to be available upon request if there are legal or ethical restrictions on sharing data publicly. For information on unacceptable data access restrictions, please see http://journals.plos.org/plosone/s/data-availability#loc-unacceptable-data-access-restrictions.

4. Your ethics statement must appear in the Methods section of your manuscript. If your ethics statement is written in any section besides the Methods, please move it to the Methods section and delete it from any other section. Please also ensure that your ethics statement is included in your manuscript, as the ethics section of your online submission will not be published alongside your manuscript.

5. Please ensure that you refer to Figure 1 in your text as, if accepted, production will need this reference to link the reader to the figure.

6. We note you have included a table to which you do not refer in the text of your manuscript. Please ensure that you refer to Table 3 in your text; if accepted, production will need this reference to link the reader to the Table.

Reviewers' comments:

Reviewer's Responses to Questions

**Comments to the Author**

1. Is the manuscript technically sound, and do the data support the conclusions?

Reviewer #1: Partly

2. Has the statistical analysis been performed appropriately and rigorously? 

Reviewer #1: No

3. Have the authors made all data underlying the findings in their manuscript fully available?

Reviewer #1: Yes

4. Is the manuscript presented in an intelligible fashion and written in standard English?

Reviewer #1: Yes

5. Review Comments to the Author

Reviewer #1: This manuscript assessed the prevalence and associated factors of adolescent fatherhood in Ethiopia. My main concern is the method the authors used to perform variable selections. In addition, it'd be interesting to include age when the first child was born to further examine whether any of the factors are associated with an earlier age of adolscent fatherhood. Below are my detailed comments.

1. Data source, line 3: please add the abbreviation "(EAs)" after "Enumeration Areas".

2. Data source, line 9: additional "," before "households".

3. Study variables, line 5: please include more details on the variables included: 1) how they were measured (e.g., how contraceptive knowledge was assessed), and 2) how these variables were coded (e.g., was age treated as continuous variable or categorical variable).

4. Data management and analysis: the authors included variables with p<0.2 from the bivariable analysis in the multivariable analysis. However, this is not the state-of-the-art approach to perform variable selection. As this study is exploratory, the authors can use models such as elastic net or lasso models to perform variable selection, along with the use of cross-validation to avoid overfitting.

6. PLOS authors have the option to publish the peer review history of their article (what does this mean?). If published, this will include your full peer review and any attached files.

Reviewer #1: No

---

## [Author Response · Author response to Decision Letter 0]

13 Oct 2020

We have prepared the manuscript based on the journal formatting style and we have revised the manuscript based on reviewer and editor comment.

---

## [Decision Letter · Decision Letter 1]

5 Nov 2020

PONE-D-20-22844R1

Prevalence and associated factors of adolescent fatherhood in Ethiopia: a multilevel analysis using the 2016 Ethiopian demographic health survey data.

PLOS ONE

Dear Dr. Misganaw Gebrie Worku,

Thank you for submitting your manuscript to PLOS ONE. After careful consideration, we feel that it has merit but does not fully meet PLOS ONE’s publication criteria as it currently stands. Therefore, we invite you to submit a revised version of the manuscript that addresses the points raised during the review process.

We look forward to receiving your revised manuscript.

Kind regards,

Yuka Kotozaki

Academic Editor

PLOS ONE

Reviewers' comments:

Reviewer's Responses to Questions

**Comments to the Author**

1. If the authors have adequately addressed your comments raised in a previous round of review and you feel that this manuscript is now acceptable for publication, you may indicate that here to bypass the “Comments to the Author” section, enter your conflict of interest statement in the “Confidential to Editor” section, and submit your "Accept" recommendation.

Reviewer #1: (No Response)

2. Is the manuscript technically sound, and do the data support the conclusions?

Reviewer #1: (No Response)

3. Has the statistical analysis been performed appropriately and rigorously? 

Reviewer #1: (No Response)

4. Have the authors made all data underlying the findings in their manuscript fully available?

Reviewer #1: (No Response)

5. Is the manuscript presented in an intelligible fashion and written in standard English?

Reviewer #1: (No Response)

6. Review Comments to the Author

Reviewer #1: The authors have addressed most of my concerns. However, I still recommend them to perform variable selections using the state-of-the-art data-driven methods. The current approach are likely to have over-fitting, which can be addressed using cross-validation.

7. PLOS authors have the option to publish the peer review history of their article (what does this mean?). If published, this will include your full peer review and any attached files.

Reviewer #1: No

---

## [Author Response · Author response to Decision Letter 1]

19 Dec 2020

according to your suggestion we have use LASSO method for the variable selection and the model with the largest adjusted R2, lowest Bayesian Information Criteria (BIC) and the smallest cross-validation prediction error were considered as the best-fitted model.

---

## [Decision Letter · Decision Letter 2]

10 Mar 2021

Prevalence and associated factors of adolescent fatherhood in Ethiopia: a multilevel analysis using the 2016 Ethiopian demographic health survey data.

PONE-D-20-22844R2

Dear Dr. Misganaw Gebrie Worku,

We’re pleased to inform you that your manuscript has been judged scientifically suitable for publication and will be formally accepted for publication once it meets all outstanding technical requirements.

Kind regards,

Yuka Kotozaki

Academic Editor

PLOS ONE

Additional Editor Comments (optional):

Reviewers' comments:

Reviewer's Responses to Questions

**Comments to the Author**

1. If the authors have adequately addressed your comments raised in a previous round of review and you feel that this manuscript is now acceptable for publication, you may indicate that here to bypass the “Comments to the Author” section, enter your conflict of interest statement in the “Confidential to Editor” section, and submit your "Accept" recommendation.

Reviewer #1: All comments have been addressed

2. Is the manuscript technically sound, and do the data support the conclusions?

Reviewer #1: Yes

3. Has the statistical analysis been performed appropriately and rigorously? 

Reviewer #1: Yes

4. Have the authors made all data underlying the findings in their manuscript fully available?

Reviewer #1: (No Response)

5. Is the manuscript presented in an intelligible fashion and written in standard English?

Reviewer #1: Yes

6. Review Comments to the Author

Reviewer #1: The authors have addressed all my concerns, and I do not have any further suggestion.

7. PLOS authors have the option to publish the peer review history of their article (what does this mean?). If published, this will include your full peer review and any attached files.

Reviewer #1: No

---

## [Editor Report · Acceptance letter]

19 Mar 2021

PONE-D-20-22844R2 

Prevalence and associated factors of adolescent fatherhood in Ethiopia: a multilevel analysis using the 2016 Ethiopian demographic health survey data. 

Dear Dr. Worku:

I'm pleased to inform you that your manuscript has been deemed suitable for publication in PLOS ONE. Congratulations! Your manuscript is now with our production department. 

Kind regards, 

on behalf of

Dr. Yuka Kotozaki 

Academic Editor

PLOS ONE